# Sparsity beyond TopK: A Novel Cosine Loss for Sparse Binary Representations

## Abstract

While binary vectorization and sparse representations have recently emerged as promising strategies for efficient vector storage and mechanistic interpretability, the integration of these two paradigms has until now remained largely unexplored. In this paper, we introduce an exciting approach for sparse binary representations, leveraging a soft TopK Cosine Loss to facilitate the transition from dense to sparse latent spaces. Unlike traditional TopK methods which impose rigid sparsity constraints, our approach naturally yields a more flexible distribution of activations, effectively capturing the varying degrees of conceptual depth present in the data. Furthermore, our cosine loss formulation inherently mitigates the emergence of inactive features, thereby eliminating the need for complex re-activation strategies prevalent in other recent works. We validate our method on a large dataset of biomedical concept embeddings, demonstrating enhanced interpretability and significant reductions in storage overhead. Our present findings highlight the clear potential of cosine-based Binary Sparsity Alignment for developing interpretable and efficient concept representations, positioning our approach as a compelling solution for applications in decision-making systems and compact vector databases.

## 1 Introduction

The rapid growth of data in various domains has necessitated the development of efficient representation techniques that not only reduce storage requirements but also enhance interpretability. Sparse representations have emerged as a promising solution, enabling the encoding of information using only a subset of active features. This sparsity leads to significant reductions in storage overhead and computational complexity, making it particularly advantageous for big data applications.

Binary representations, characterized by their compactness and efficiency, have also gained traction in recent years. By representing data with binary 1-bit values, models can achieve faster inference times and lower memory footprints, which are crucial for deployment in resource-constrained environments. However, the integration of sparsity and binary representation remains unexplored.

Current strategies for producing binary representations rely on quantization techniques that convert continuous values into discrete levels. While quantization can yield compact representations, it struggles to effectively capture the nuanced activation patterns inherent in many linear autoencoders used for sparsification. Indeed, features of sparse representations trained with existing techniques typically exhibit varying degrees of activation, reflecting the complexity of the underlying data, as detailed in section 2. This variability poses a significant challenge for traditional quantization methods, which tend to impose rigid thresholds that fail to accommodate gradual transitions in feature activations, as well as the varying information needs of different concepts to be embedded. Consequently, the resulting binary representations often lack the fidelity necessary for accurate reconstruction and interpretation.

Despite the advantages of both sparse and binary representations, there is currently no established methodology for generating sparse binary embeddings that leverage the strengths of both paradigms. In this paper, we introduce a novel approach—a soft cosine alignment loss for Binary Sparsity Alignment (BSA)—that addresses this gap by enabling the effective transition from dense to sparse binary latent spaces. Our method preserves the interpretability and efficiency of sparse representations, without encouraging intensity variations in feature activation by applying binarization at training time, thereby paving the way for future advancements in representation learning.

## 2 RELATED WORKS

### 2.1 SPARSE EMBEDDINGS

Sparse embeddings are a form of vector data representation where the majority of the components are zero, significantly reducing the dimensionality of the data while retaining essential information.

Sparse embeddings offer several key advantages, particularly in terms of computational efficiency and scalability. These embeddings can help reduce the effective dimensionality of data, leading to faster computations and lower storage requirements, which is especially beneficial in large-scale applications (Nguyen et al., 2012; Liang et al., 2021). Sparse embeddings also allow for efficient data engineering and transfer learning, as they combine the strengths of high-dimensional linear models with the benefits of latent factor representations (Van Balen & Goethals, 2021). Moreover, the efficiency of sparse embeddings is further enhanced by the reduced number of floating-point operations due to sparse matrix multiplications, leading to quicker retrieval tasks (Paria et al., 2020).

In addition to computational benefits, sparse embeddings improve model interpretability and robustness. They provide better interpretability compared to dense embeddings, as seen in various applications from word embeddings to retrieval tasks (Sun et al., 2016; Kong et al., 2023). The sparse structure also enhances robustness to noise and interference, maintaining competitive accuracy while offering advantages in hyperparameter tuning and learning speed (Ahmad & Scheinkman, 2019). Furthermore, sparse embeddings demonstrate superior performance in capturing meaningful data structures and achieving high accuracy in tasks like signal recovery and object classification, often outperforming other competitive algorithms (Nguyen et al., 2012; Medini et al., 2021). Overall, these attributes make sparse embeddings a valuable tool in machine learning and data processing, providing both practical and theoretical benefits.

Sparse representations in machine learning have been extensively studied, with various methods proposed to enforce sparsity constraints. A notable technique is top-k sparsity, which has been successfully applied in k-sparse autoencoders. These autoencoders retain only the k highest activations in the hidden layers, leading to improved classification performance, while maintaining simplicity and fast encoding stages (Makhzani & Frey, 2014). Autoencoders compress input data into a compact internal representation and then reconstruct it, aiming to minimize the difference between the input and output in order to learn efficient data representations through unsupervised learning (Bisong, 2019; Bank et al., 2021). Modern autoencoders often consist in shallow linear systems, even in advanced mechanistic interpertability use cases (Gao et al., 2024; Templeton et al., 2024).

Similarly to k-sparse autoencoders, winner-take-all autoencoders employ a competitive mechanism that ensures only the most significant activations are preserved, facilitating the learning of hierarchical and deep sparse representations in an unsupervised manner (Makhzani & Frey, 2015).

Another prominent approach involves L1 and L2 regularization. Older techniques, such as Sparse LSA, leverage L1 regularization to enforce sparsity on the projection matrix, yielding compact and interpretable topic-word representations (Chen et al., 2011). More recently, L1-regularized autoencoders have demonstrated the ability to achieve high compression ratios with minimal artifacts (Chung et al., 2024). For topic modeling, L2 regularization has been employed to balance the sparsity of topic and word distributions, optimizing the trade-off between sparse representation and model accuracy (Anon, 2018). Additionally, non-smooth L1 regularization has been shown to be more effective than smooth approximations in training sparse autoencoders, directly targeting the sparsity objective (Amini et al., 2022).

Anchor-based transformation methods have also gained traction. These projection methods introduce a small set of anchor embeddings and a sparse transformation matrix to efficiently represent large vocabularies, enhancing scalability and performance in tasks such as text classification and language modeling (Liang et al., 2021; Chen et al., 2019; Medini et al., 2021). These methods overcome the scalability issues of embedding large vocabularies by capturing the underlying structure and similarities between objects more efficiently than traditional independent embedding techniques.

However, existing sparse methods generated real-valued embeddings, which might not be sufficient to enable efficient search and storage, and are less suitable as input to neurosymbolic approaches than binary embeddings.

### 2.1.1 BINARY EMBEDDINGS

Binary embeddings are a nonlinear dimension reduction technique that transforms high-dimensional data into binary strings while preserving the structure of the original space (Yi et al., 2015). They can for instance be used to convert real-valued embeddings into binary representations (Sherki et al., 2021). Binary embeddings significantly reduce the size of real-valued embeddings. Tissier et al. (2019) note that they can achieve a 97% reduction in vector size compared to real-valued embeddings, and they enable much faster vector operations compared to real-valued embeddings. Indeed, vector operations can be performed using bitwise operators instead of floating-point arithmetic, Top-K queries can be executed up to 30 times faster with binary vectors compared to real-valued vectors, and loading binary vectors from storage is also much quicker. Despite the significant reduction in size, binary embeddings can retain most of the semantic information present in the real-valued ones.

Recent research has explored various approaches to train binary embeddings for efficient representation and retrieval. Huynh & Saab (2020) proposed fast binary embeddings using quantized Johnson-Lindenstrauss transforms, achieving polynomial and exponential error decay rates. Zhuang et al. (2016) developed a triplet-based deep binary embedding network, formulating the problem as multi-label classification and significantly reducing training time. Mostard et al. (2022) presented a semantic-preserving siamese autoencoder for quantizing word embeddings, preserving semantic information while reducing dimensionality. Their approach outperformed baselines on word similarity and sentence classification tasks, with individual bits holding interpretable semantic information.

However, these methods do not attempt to produce sparse binary representations, and their application to already-sparse embeddings offers no guarantee of perserved sparsity. The aim of this work is to propose a method which enables simultaneous sparsification and binarization of the input data, with stable and intuitive learning dynamics.

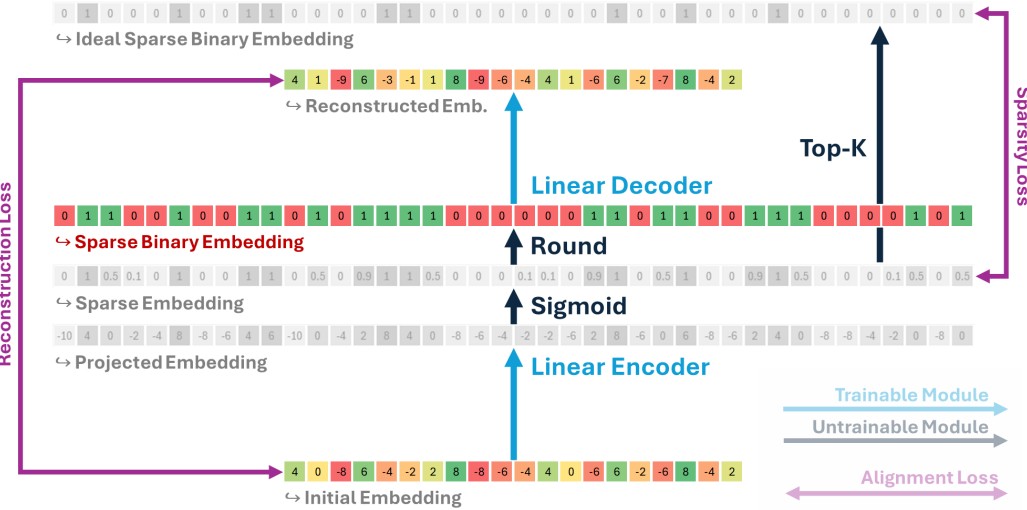

Figure 1: **Overview of our soft cosine-based Binary Sparsity Alignment approach.** This figure illustrates the process of generating sparse binary embeddings from a given dense embedding using a linear auto-encoder pipeline, as well as the training losses applied to this system. The given initial embedding is first projected to dimension $N$ using a linear encoder, after which a sigmoid activation function is applied to obtain the sparse embedding (with all activations ranging from 0 to 1). This sparse but non-binary embedding is then then rounded to produce the sparse binary embedding. The sparse binary embedding is passed through a linear decoder to reconstruct the initial embedding as accurately as possible. The reconstruction alignment is enforced by minimizing the cosine distance between the initial embedding and the reconstructed embedding. The sparsity alignment is achieved by minimizing the cosine distance between the sparse embedding and an ideal sparse binary embedding obtained after applying a TopK operation to the non-binary sparse embedding. The training loss is the sum of the reconstruction and sparsity alignment losses.

## 3 METHODOLOGY

### 3.1 BINARY SPARSITY ALIGNMENT (BSA)

Our method is applicable to the binary sparsification of latent vectors derived from models trained under a contrastive learning objective, and whose relatedness can be measured using the cosine similarity metric. Contrastive learning and the cosine similarity metric in particular have become the standard unsupervised representation learning toolbox for years, with few challengers so far, and they are thus well-studied at this point (Oord et al., 2018).

Our approach can generate sparse embeddings of size $N$ and average feature activation count $K$, capable of reconstructing the initial normalized embeddings of size $E$ using a single linear decoder.

Similarly to the previously-mentioned sparsification approaches, we train our sparse binary encoder using an autoencoder pipeline ($E \to N \to E$). To achieve binary activations, we apply the sigmoid activation function after the linear layer of the encoder, to bring activations in the 0-to-1 range.

$$\textbf{SparseEmbedding} := \text{Sigmoid}(\text{LinearEncoder}(\text{InitialEmbedding}))$$

By using small initial weights, and because of the isotropic nature of contrastive embedding spaces, our latent embeddings at initialization consist mostly of values just above or below the natural center of 0.5 after application of the sigmoid activation, with roughly half of them above the threshold.

Rounding is subsequently applied to binarize the features during training, before passing them to the decoder layer. As rounding is not a differentiable operation, we define a passthrough rounding function in PyTorch for this purpose, which we describe in subsection 3.3.

$$\textbf{SparseBinaryEmbedding} := \text{PassthroughRound}(\text{SparseEmbedding})$$

The key intuition behind our approach is our framing of the desire for sparsity not as a constraint (e.g. TopK) nor as a general regularization (e.g. L1), but instead as an alignment problem. Sparsity alignment can be measured as the cosine similarity between the currently-produced latent vectors and their corresponding ideal sparse vectors, with their $K$ most activated features all positive and with values close to 1 and their $N - K$ unactivated features close to 0.[1]

$$\textbf{SparsityAlignmentLoss} := \text{CosineDistance}($$
$$\text{SparseEmbedding},$$
$$\text{TopK}(\text{SparseEmbedding})$$
$$)$$

Framing the sparsity desire as cosine alignment plays very well with our autoencoder reconstruction loss, also framed as a cosine distance minimization between initial and reconstructed embeddings.

$$\textbf{ReconstructionLoss} := \text{CosineDistance}($$
$$\text{LinearDecoder}(\text{SparseBinaryEmbedding}),$$
$$\text{InitialEmbedding}$$
$$)$$

The bounded nature of the cosine distance metric (0-2) ensures that both losses evolve in similar value ranges, and can meaningfully be combined, unlike MSE distances which suffer from explosion effects at large $N$ values. The sum of these two losses (possibly weighted as desired by modulating hyperparameter $\alpha$) forms the total loss of the system, with no need for any other regularization or feature re-activation term.

$$\textbf{TrainingLoss} := \text{ReconstructionLoss} + \text{SparsityAlignmentLoss} * \alpha$$

---

[1]The degree of rejection of non-zero values for unactivated features can be controlled with a shift term substracted to this ideal vector, by replacing the TopK(...) term with TopK(...) $- \varepsilon$, although this is not necessary.

### 3.2 POSSIBLE TOPK-FOCUSED LOSS

In case where flexibility in $K$ values is not desired, a third loss term can be added, measuring the reconstruction loss between LinearDecoder(TopK(SparseEmbedding)) and the initial embedding. This loss term is very similar to the rounded reconstruction loss, but ensures the maximization of information in the top $K$ features of the embedding, without respect for the rounding threshold.

$$\textbf{FixedKReconstructionLoss} := \text{CosineDistance}($$
$$\text{LinearDecoder}(\text{TopK}(\text{SparseEmbedding})),$$
$$\text{InitialEmbedding}$$
$$)$$

We envision but do not evaluate in this study that several such losses with varying $K'$ values could be used to train Matrioshka-style sparse binary embeddings (Kusupati et al., 2022). However, it might also be possible to achieve this by sorting the activated features by the norm of their associated decoder weight without requiring to add additional loss terms. As time was insufficient to compare these two approaches meaningfully, we leave such analysis for future work.

### 3.3 PASS-THROUGH ROUND FUNCTION

As noted previously, our reconstruction loss is calculated on the basis of the rounded (binarized) sparse embedding. This poses a significant challenge as the Round function produces no gradient for its input, which means that the sparse embedding would receive no training signal from the reconstruction loss, which cannot possibly lead to good outcomes.

To overcome this problem, we define a pass-through $\text{Round}(x)$ function which propagates its gradient to $x$. This is a strategy similar to the one used in quantization/codebook strategies already used in other works (Baevski et al., 2020; Esser et al., 2021).

$$\textbf{PassthroughRound}(x) := \text{Round}(\text{NoGrad}(x)) + (x - \text{NoGrad}(x)))$$

This function behaves like $\text{Round}(x)$ at inference, but like $x$ during gradient descent. The effect of this pass-through will be to send a positive signal to features of $x$ whose activation would positively benefit the final reconstruction, and a negative signal to features whose activation has negatively affected the reconstruction (something that is further substantiated below).

### 3.4 TRAINING DYNAMICS

This section presents an analysis of the training dynamics of our approach, to build up an intuitive understanding of its underlying mechanisms. Its aim is not to depict in a completely accurate way the derivatives, but to outline the main components and their impact. We also compare the resulting dynamics with those of the popular TopK and L1 approaches, to highlight perceived improvements over the state of the art.

**Let us first examine the reconstruction loss.** Because rounding is already applied during training, our approach does not suffer from any mismatch between inference and training times. The value vectors from the decoder do not have the possibility to be modulated, leading to a lesser risk of "weak activation" of the latent features.

Unlike the frequently used TopK clipping method, our approach also propagates the gradient to the entire feature space, and not only its first K components; this clipping is a key weakness of TopK training which give no chance to weakly activated features to reactivate by receiving a positive signal once they fall outside the top K activated features.

Finally, the optional TopK loss term can be used to favor a representation of acceptable accuracy even for concepts activating more than K features, ensuring that our method can be used as a drop-in replacement in existing TopK pipelines.

**Let us now consider the sparsity alignment loss.** Ignoring the effects of normalization for a moment, the general dynamic of cosine similarity is that it increases when features have the same polarity (sign) in both compared vectors, and decreases sharply when features have opposite polarity in the compared vectors. Given our feature activations can only be positive and between 0 and 1, the dynamic is better understood in the setting where these vectors are shifted downwards by the $\varepsilon$ term, with values between $-\varepsilon$ and $1 - \varepsilon$ (although the dynamic applies identically for $\varepsilon = 0$).

In this setting, the top $K$ features of the sparse vector will receive a positive gradient towards their maximal value $(1 - \varepsilon)$, while the remaining $N - K$ features will receive a negative gradient towards their minimal value $(-\varepsilon)$. This gradient will then be counter-balanced by the reconstruction loss, where features contributing positively to the reconstruction might or might not get deactivated depending on the weight $\alpha$ assigned to the alignment loss.

This approach possess a significant advantage over a classical L1 loss, because only the bottom $N - K$ features will receive a downward push, while the top $K$ features will be boosted. This makes the training significantly more stable and less prone to total collapse (where significantly fewer than $K$ features remain activated on average).

Compared to TopK clipping, this strategy has the additional benefit of encouraging sparsity towards a specific sparsity goal without enforcing an identical number of activation for every concept. Indeed, concepts whose representation requires more than $K$ active features to be accurate will likely end up with a larger decrease in reconstruction loss when these features remain activated, than the increase of sparsity alignment loss that they cause. By modulating $\alpha$, it is possible to enforce more or less the sparsity constraint, for instance by increasing its importance as the training progresses.

## 4 EXPERIMENTAL SETUP

We validate our approach on a dataset of biomedical concept embeddings produced using a semantic model trained contrastively for the biomedical domain, called BioLORD-2023-C (Remy et al., 2024). Our concept list orginates from UMLS (Bodenreider, 2004) and contains more than 4M unique concepts, referenced by their canonical name (synonyms not being used in this setup).

Note that the embedding model is not finetuned in this experiment and considered as a gold standard, since our objective is to sparsify already-existing concept embeddings while increasing their interpretability. We leave as future work whether our losses can be used to train sparse binary features in an end-to-end manner.

We aim to answer four research questions about Binary Sparsity Alignment (BSA) in this paper:

**RQ1:** **Can dense embeddings be mapped to sparse binary embeddings without significant information loss using BSA?** Here, we define *significant information loss* as an average reconstruction loss above the cosine margin between different concepts used while training the biomedical semantic model ($\mu = 0.15$), as this would lead to frequent concept mix-ups.

**RQ2:** **Which role play hyperparameters $N$ and $K$ in the reconstruction capabilities of BSA?** In particular, it it possible to reduce the reconstruction loss to an arbitrary low value by increasing $N$, at which rate, and how should $K$ be adapted as $N$ increases.

**RQ3:** **Is BSA sufficient to achieve any desired degree of sparsity?** Here, we define *achieving a desired degree of sparsity* as producing on average $K$ activated features per concept, with a standard deviation significantly inferior to $K$ itself (for any reasonable value of $K$).

**RQ4:** **What does the activation pattern of the latent features look like after BSA training?** In particular, can we detect the presence of a significant number of "dead" latent features which do not activate for any known concept? Are all features activating a roughly equal number of times, or are there features shared among more concepts than others?

To answer these research questions, we train 16 autoencoders with BSAs of varying hyperparameters ($N \in \{4, 8, 16, 32\} * 1024$, $K \in \{32, 64, 128, 256\}$, $\alpha = 1$, $\varepsilon = 0.125$) and measure their ability to sparsify the latent space with minimal reconstruction loss.

We also analyze the hyperparameter sensitivity of the approach.

# 5 EXPERIMENTAL RESULTS

## 5.1 RECONSTRUCTION ALIGNMENT

In all 16 experiments, the average reconstruction loss was significantly inferior to the angular margin enforced on the input data, signifying that the sparse binary embeddings produced by the approach were precise enough to keep concepts distant from each other the vast majority of the time.

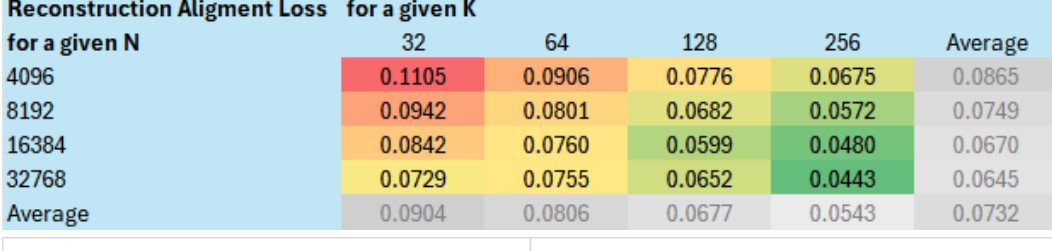

| Reconstruction Aligment Loss | for a given K | | | | |
| --- | --- | --- | --- | --- | --- |
| for a given N | 32 | 64 | 128 | 256 | Average |
| 4096 | 0.1105 | 0.0906 | 0.0776 | 0.0675 | 0.0865 |
| 8192 | 0.0942 | 0.0801 | 0.0682 | 0.0572 | 0.0749 |
| 16384 | 0.0842 | 0.0760 | 0.0599 | 0.0480 | 0.0670 |
| 32768 | 0.0729 | 0.0755 | 0.0652 | 0.0443 | 0.0645 |
| Average | 0.0904 | 0.0806 | 0.0677 | 0.0543 | 0.0732 |

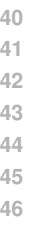
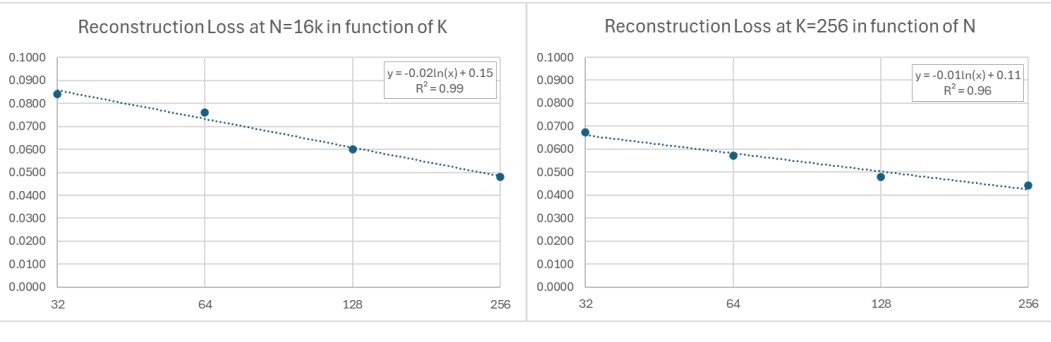

Table 1: Average Reconstruction Loss after binary sparsification for a given N and K.

As expected, increasing either the embedding dimension $N$ or the activation count $K$ increases the precision of the reconstruction. Doubling $K$ decreases the loss by about 0.175 on average, while doubling $N$ decreases it by about 0.1 on average. From an information theory point of view, the total number of bits necessary to represent the vector corresponds to $K \log_2(N)$, indicating that doubling $N$ is more effective than doubling $K$. However, doubling $N$ eventually stops being effective (and might even be detrimental), which is particularly visible for the last row where $N$ is set to 32768.

We can estimate the achieved compression ratio using the above formula. Storing our 128-sparse binary embeddings of size 16384 requires 128*14 bits, or 1792 bits. This can be compared to the original dense embeddings, which required 768*32 bits, or 24576 bits (14 times more).

Similar findings hold true in the case where the top $K$ features are activated instead of all features rounding to 1, to the exception of values of $K$ significantly inferior to $N$, where it seems that sparsity alignment was not successfully achieved (we will analyze this further in subsection 5.2).

| Reconstruction Aligment Loss | for a given K | | | | |
| --- | --- | --- | --- | --- | --- |
| for a given N | 32 | 64 | 128 | 256 | Average |
| 4096 | 0.1139 | 0.0919 | 0.0786 | 0.0701 | 0.0886 |
| 8192 | 0.1052 | 0.0851 | 0.0690 | 0.0588 | 0.0795 |
| 16384 | 0.1085 | 0.0854 | 0.0614 | 0.0484 | 0.0759 |
| 32768 | 0.1374 | 0.1130 | 0.0832 | 0.0461 | 0.0949 |
| Average | 0.1163 | 0.0938 | 0.0731 | 0.0558 | 0.0848 |

Table 2: Average Fixed-TopK Reconstruction Loss after binary sparsification for a given N and K.

To stay out of the regime where the top $K$ features always accurately represent the initial embedding, increasing $N$ and $K$ simultaneously appears necessary. When the desired sparsity level $K/N$ drops below 0.75%, the model appears incapable of achieving accurate TopK reconstruction, likely because it requires more than K features to achieve a good reconstruction alignment.

## 5.2 SPARSITY ALIGNMENT

In addition to the the reconstruction results achieved by our proposed approach, its suitability for binary sparsification also require the desired sparsity levels to be achieved, something we have not verified until now. We do so by computing the average number of activated features per concept, as well as its standard deviation.

| Average of Activated Features for a given N | for a given K 32 | 64 | 128 | 256 | Average |
|---|---|---|---|---|---|
| 4096 | 34.91 | 64.78 | 128.96 | 260.57 | 122.30 |
| 8192 | 37.30 | 66.87 | 128.91 | 258.35 | 122.86 |
| 16384 | 49.13 | 72.52 | 132.00 | 256.70 | 127.59 |
| 32768 | 82.91 | 101.39 | 154.00 | 259.48 | 149.45 |
| Average | 51.07 | 76.39 | 135.97 | 258.77 | 130.55 |

Table 3: Average number of activated features, after binary sparsification, for a given $N$ and $K$.

As suggested by the misalignment between the natural and top-k reconstruction losses reported in subsection 5.1, extreme sparsity levels are not achievable using the chosen hyperparameters. While the sparsity alignment loss could be increased to enforce the desired sparsity level more strongly, this would likely be counter-productive, suggesting that decreasing $N$ to maintain a reasonable sparsity level would achieve better outcomes.

However, in all cases, the standard deviation of activation count was very low, confirming that while some concepts activate more features than others, the number of activations usually stays within a reasonable range around the target $K$. This can be confirmed by taking the average batch minimum and maximum count of activations, which are as expected well within 3 standard deviations (e.g. with a minimum of 240 and a maximum 280 for K=256, considering a batch size of 256).

| Std of Activated Features for a given N | for a given K 32 | 64 | 128 | 256 | Average |
|---|---|---|---|---|---|
| 4096 | 5.07 | 5.08 | 5.34 | 9.07 | 6.14 |
| 8192 | 6.62 | 5.93 | 5.86 | 9.70 | 7.03 |
| 16384 | 15.14 | 10.29 | 9.23 | 9.30 | 10.99 |
| 32768 | 38.54 | 26.58 | 19.39 | 9.16 | 23.42 |
| Average | 16.34 | 11.97 | 9.96 | 9.31 | 11.89 |

Table 4: Standard deviation of number of activated features, as Table 3, for a given $N$ and $K$.

Overall, our experiments indicate that a large range of desired compression and sparsity levels are achievable with our approach, without requiring hard constraints.

## 5.3 ACTIVATION PATTERNS

A common failure mode of sparse autoencoders is their tendency to under-use the feature space by leaving encoder features which never activate, sometime referred to as "dead latents". Most papers cited in the Related Works section resort to re-activation strategies to mitigate this issue, encouraging the activation of features whose activation was not witnessed recently.

These strategies are not necessary for avoiding permanently-inactive features with our approach. To demonstrate this, we observe the distribution of feature activations in one our trained autoencoders on a test set of 775k concept names extracted from SnomedCT (Schulz & Klein, 2008).

Figure 2 demonstrates that not only the activation count per concept remains in distribution (left), but also that all features were activated a reasonable amount of time (right). 99% of all features activated between 205 and 165673 times (between 0.03% of concepts and 23.4% of embedded concepts).

This figure is not sensitive to the 0.5 threshold used for rounding feature activations, as the vast majority of feature activations offer a pretty clear signal. An astonishing 99.95% of non-binary activation intensities are beyond the central range between 0.1 and 0.9 (although this is partly because of the high sparsity of the embeddings), and 86% of activated features have an non-binary activation intensity above 0.6 (see Figure 3).

Overall, these results demonstrate the outstanding stability and soundness of the activation patterns of our trained sparse encoder, without the need for re-activation strategies.

# 6 CONCLUSION

In this work, we introduced our soft cosine-based Binary Sparsity Alignment approach, a novel 1-bit sparsification method capable of bridging the gap between dense and sparse binary latent spaces. Our experiments demonstrated that BSA transforms dense embeddings into interpretable sparse binary representations with minimal information loss, thanks to its soft cosine sparsity alignment loss. Unlike existing approaches that impose rigid sparsity constraints or suffer from unstable training dynamics, sparsity alignment-based training is stable and flexible by design. Its framework not only provides practical benefits but also establishes a theoretical foundation for understanding binary sparsification in the context of contrastive learning. Overall, our findings underscore the effectiveness and versatility of soft sparsity alignment, paving the way for future advancements in efficient representation storage and more interpretable decision-support systems.

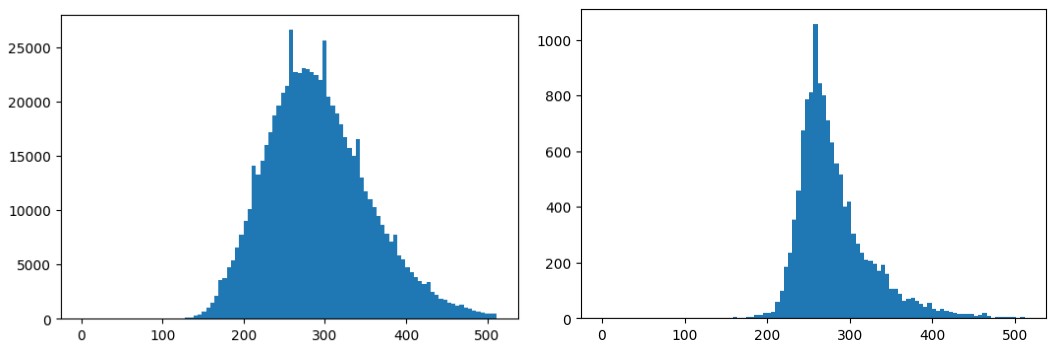

Figure 2: Feature Activation Patterns on a SnomedCT test set of 770k concept names.
Left: Distribution of concept names by feature activation count.
Right: Distribution of features by feature activation count.

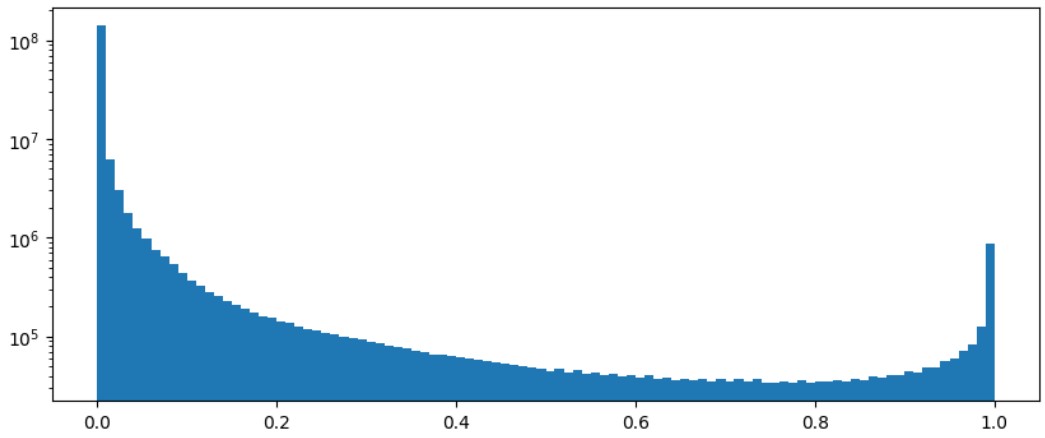

Figure 3: Distribution of Non-binary Feature Activation Intensities on the SnomedCT test set.

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
