# OpenReview forum: "Sparsity beyond TopK: A Novel Cosine Loss for Sparse Binary Representations"
_ICLR.cc/2025/Conference — ICLR 2025 Conference Withdrawn Submission_

### Official Review · Reviewer_TR2B · 2024-10-19

**Soundness:** 1
**Presentation:** 1
**Contribution:** 1
**Rating:** 1
**Confidence:** 4

**Summary:**

The paper presents a method to produce sparse binary embeddings based on a cosine similarity to a top-K sparse embedding. The authors expect that this will allow them to have a soft-sparsity constraint that does not unnecessarily penalize features that might appear in the data. They demonstrate the effectiveness of their sparse binary embedding in an autoencoder. They linearly project the data in a latent space, then apply a signmoid and a rounding to binary outputs, a second  representation were only the top-k values of the sigmoid is set to one and the rest to zero is also returned. Then they use a cosine metric between the output of the sigmoid and the top-k representations, to obtain something like aa soft-top-k penalty for representation. Then they reconstruct the input from that latent space with a linear decoder and optimise the encoder and decoder (both linear) with a a cosine loss between input and reconstruction.

The method is sensible, although limited in scope - addressing only the issue where top-k would ignore significant features. I am not sure how often that is a problem with top-k sparse representations. Also, if you want to get soft approximations of sparse binary latents, I would expect to see something based on l1 sparsity penalty. I think Mairal and Bach, and the community around them,  about 10-15 years ago did a lot of work around demonstrating that the l1 is convex relaxation of binary sparse coding, and therefore yields an optimal solution. I think there are still open problems with l1 sparse coding, e.g. how do you identify optimal hyper parameters, how do you find an optimal subgradient and things like that. However, that literature around l1 is barely discussed in the paper.

The major problem with the paper is that it proposes a new method, albeit somewhat incremental, and it does not provide an analytical proof of any interesting property nor do the  authors demonstrate its efficacy in an application. Instead only some numerical experiments that resemble sanity checks are presented. For instance that after training the top-k and the binary sparse representation are correlated. This is rather obvious since their cosine similarity was part of the objective function to be optimised.

I would strongly recommend that the authors study the sparse coding literature in greater depth and/or perform more experiments on realistic data with reasonable numerical targets for method validation.

Also, some typos here and there:
Line 309: seems syntactically wrong
Line 281: This approach possesses (not possess)
Line 84: Modern auto encoders often consist of (not in)

**Strengths:**

New and relatively sensible method. The implementation has been validated with numerical experiments that demonstrate the model works as designed.

**Weaknesses:**

The paper implies ignorance of a large part of related literature. Going for a cosine similarity between a binary vector with no constraints and a binary vector with top-k constraint is a strange way to relax the hard sparsity condition. The numerical experiments only validate proper function of the implementation, they don't demonstrate an application to a real problem. Also, there is no comparison with alternative sparsity methods

**Questions:**

Why did you chose to work with a cosine similarity with top-k rather than a norm penalty, e.g. l_1?
Can you compare your work with some other form to sparsify latent representations?
How would your representation work in a real problem, often targeted with sparse representations, e.g. denoising?

---

### Official Review · Reviewer_KPj6 · 2024-10-31

**Soundness:** 2
**Presentation:** 1
**Contribution:** 1
**Rating:** 1
**Confidence:** 3

**Summary:**

This paper introduces Binary Sparsity Alignment (BSA), a novel method for converting dense embeddings into sparse binary representations using a cosine-based loss function. Unlike traditional approaches that use hard constraints or regularization, BSA treats sparsity as an alignment problem, leading to more stable training and better preservation of semantic information. Testing on biomedical concept embeddings demonstrated that the method achieves 14x compression while maintaining semantic relationships and avoiding common issues like "dead features," all without requiring special feature reactivation strategies that are typically needed in other approaches.

**Strengths:**

Novel cosine-based loss function for simultaneous sparsification and binarization.

Achieves significant compression (14x reduction).

Stable training without dead features.

Tested successfully on biomedical concept embeddings.

**Weaknesses:**

There is no evaluation of sparse binary embeddings on downstream tasks regarding performance. At this moment, it is hard to even tell whether the compressed embeddings work. Even worse, there is no baseline comparison.

The results were reported based on only one domain (biomedical concepts).

There is no analysis of training time or computational requirements.

There is no ablation study on hyperparameters like the loss weighting parameter, batch size etc.

The presentation in this paper is poor, the authors did not even write the table in latex, but screenshot some excel sheets.

**Questions:**

It is unclear how the approach would scale to even larger dimensions?

---

### Official Review · Reviewer_pLMZ · 2024-11-02

**Soundness:** 2
**Presentation:** 3
**Contribution:** 2
**Rating:** 3
**Confidence:** 3

**Summary:**

- The paper introduces a novel approach called Binary Sparsity Alignment (BSA) that transitions dense embeddings to sparse binary representations using a soft TopK Cosine Loss, enhancing interpretability and reducing storage needs. BSA avoids rigid sparsity constraints and unstable training dynamics, offering a stable and flexible method for sparse binary encoding.

**Strengths:**

- The research focuses on a narrow topic, specifically Binary Sparsity Alignment, which is rarely addressed by others. This method aims to enhance interpretability while reducing storage requirements.
- The approach is validated using a biomedical dataset, demonstrating effective sparsification with minimal information loss.

**Weaknesses:**

- The authors cite the paper on contrastive learning of predictive coding; however, the relationship between their work and the contrastive learning perspective (Oord et al., 2018) is not thoroughly discussed.
- Many of the equations presented are not commonly found in other literature, making it challenging for readers to grasp the intended message. The organization of the paper could benefit from improvement.
- The experimental validation is insufficient to support the proposed ideas. Additional experiments and the inclusion of datasets beyond the biomedical context are necessary for a more comprehensive evaluation.

**Questions:**

- How can your study benefit efficient storage and fast inference? In what situations and environments would this be applicable?
- How does your method enhance interpretability, especially in fields like biomedical concept embeddings, where understanding the underlying data is crucial?
- How can you demonstrate practical benefits while also establishing a theoretical foundation for understanding binary sparsification in the context of contrastive learning? Can you explain this more simply?

---

### Author Response · Authors · 2024-11-26
**Further guidance welcome :)**

Dear Reviewers,

First, I would like to sincerely thank each of you for dedicating your time to review my paper. I greatly appreciate your thoughtful feedback, and I recognize that your comments will be invaluable as colleagues and myself work to improve this research. This is my first submission in this field, and as a recent PhD graduate in NLP, I was aware that the chances of acceptance were going to be low. However, my primary goal was to receive constructive feedback and gauge interest in the proposal. Your insights are already helping me refine my approach and consider important next steps for advancing the work. I still think that there is tremendous value unlockable on this research track, and I didn't want this idea to die in a drawer after I moved on from my PhD.

I understand that the current version of the paper lacks the depth of experimentation and clarity necessary for a stronger contribution, and I fully agree with your assessments on the matter. I also realize that the paper cannot be "saved" in the short time frame of two weeks, as additional experiments and improvements to the presentation are necessary to make the work more compelling. Moving forward, I would like to initiate a discussion about the specific changes and research directions that would make this paper more interesting in your eyes. Would you recommend offering this line of work to a colleague?

For example, I hear that additional experiments beyond the biomedical domain would be extremely helpful. I would greatly appreciate any suggestions for other datasets or applications where the method could be tested, especially in real-world contexts. Also, I understand that a more comprehensive comparison to other sparsification methods, such as L1 regularization or other relevant approaches, is necessary. If you have any recommendations for related papers or follow-up works, I would be grateful for the guidance to help position this paper more effectively within the broader literature.

On the topic of sparsification, I would like to specifically address Reviewer TR2B’s suggestion that L1 regularization is the optimal solution for sparsity. While L1 sparsity is indeed well-established, it is important to note that L1 is a continuous loss, and it has been shown to fail at large scales. This is part of the reason why recent works, such as those from OpenAI and Anthropic in mechanistic interpretability, do not rely on L1 in the way that earlier methods did. I will refer to those papers for explanation as to why.

Additionnally, the binary nature of the embeddings I propose introduces challenges that L1 does not address effectively. Firstly, L1 regularization does not align with the binary structure of the representations. Secondly, applying L1 regularization in the context of a cosine similarity loss as reconstruction loss does not produce satisfactory results. Cosine similarity loss operates within a limited -1 to +1 range, while L1 regularization scales unreasonably high when applied to embeddings with large dimensions. In contrast, the binary embeddings in our approach are generated through end-to-end training, where the loss is computed on an already-binarized representations. This ensures that there is no mismatch between the training and inference objectives.

Moreover, when L1 is paired with an MSE loss (which is typically used for large-dimensional embeddings), the result is ineffective at semantic compression, as MSE treats amplitude as important, while in our case, cosine similarity is optimized to preserve the semantic structure rather than the amplitude of the embeddings. Thus, the primary contribution of our method is the use of a cosine-based loss that naturally aligns with the binary structure, preserving the semantic information without relying on the drawbacks of L1+MSE.

I would welcome further feedback on this argument and the rationale behind our choice of a cosine-based loss. Additionally, if there are other sparsification techniques beyond TopK and L1+MSE that you believe should be compared with our method, I would be eager to hear about those as well.

Once again, thank you for your constructive feedback. I look forward to any further suggestions you may have on how this work can be enhanced and better positioned for future submissions. Your advice on datasets, related research, and potential follow-up experiments will be extremely helpful as me or a colleague continue to develop this idea.

Best regards,
The Author

---

### Note · Authors · 2024-12-03

I have read and agree with the venue's withdrawal policy on behalf of myself and my co-authors.